# Cultural adaptation and psychometric evaluation of Childbirth Experience Questionnaire 2 in Karnataka state, India

Paridhi Jha[1], Vikas Kumar Jha[2], Bharati Sharma[3], Ajeya Jha[4], Kerstin Erlandsson[5], Malin Bogren[6]*

1 Foundation for Research in Health Systems, Bangalore, Karnataka, India, 2 Indian Institute of Technology Delhi, New Delhi, India, 3 Indian Institute of Public Health Gandhinagar, Gandhinagar, Gujarat, India, 4 Department of Management Studies, Sikkim Manipal Institute of Technology, Rangpo, Sikkim, India, 5 School of Health and Welfare, Dalarna University, Falun, Sweden, 6 Institute of Health and Care Sciences, Sahlgrenska Academy, University of Gothenburg, Gothenburg, Sweden

* malin.bogren@gu.se

**Data Availability Statement:** The minimal dataset cannot be uploaded to any server accessible from outside India as per the Indian Council for Medical

## Abstract

### Background

Women's birthing experience is a sensitive indicator of the quality of childbirth care and can impact the physical and mental health of both women and their neonates. Negligible evidence exists on Indian women's birth experiences and–to the best of authors' knowledge–no questionnaire has been tested in India for measuring women's birthing experiences. This study aimed to test the construct validity and reliability of the *Kannada*-translated Revised Childbirth Experience Questionnaire.

### Methodology

A cross-sectional survey was carried out among postnatal women (n = 251, up to six months postpartum, with a live healthy neonate) who had given birth at a public or private health facility using the Kannada-translated CEQ2 in two districts of Karnataka. Data were collected at participants' homes after seeking written informed consent. Model fit was determined by Confirmatory Factor Analyses.

### Results

The 4-factor model of the CEQ2 showed good fit after deletion of one item (item 8, subcategory "participation") with CMIN = 1.33; SRMR = 0.04; GFI = 0.92, CFI = 0.98, TLI = 0.99, RMSEA = 0.037 and p value 0.002). The Cronbach alpha values were acceptable for the four subscales (0.92, 0.93, 0.97, 0.91) as well as for the overall 21-item scale (0.84).

### Conclusions

The *Kannada*-translated CEQ2 is a reliable tool to measure the childbirth experiences among *Kannada*-speaking women and can serve as a reliable ongoing evaluation of women's birth experiences.

Research Ethical Guidelines as the data contain sensitive patient information. However, data can be accessible upon reasonable request. Request can be sent to: Chairperson, Ethical Review Board, Foundation for Research in Health Systems Indian Society for Health Administrators Building, Door Number 3009, II-A Main, 17th Cross, Banashankari II Stage, K. R. Road, Bengaluru-560070. India Landline (between 10 am- 6 pm IST on working days) +91 80 2657 7978 Email ID: admin@frhsindia.org.

**Funding:** This study was conducted with financial assistance from Aastrika(AastarUrmika Health Systems Pvt. Ltd.) https://www.aastrika.org/ The funders had no role in study design, data collection and analysis, decision to publish, or preparation of the manuscript.

**Competing interests:** The authors have declared that no competing interests exist.

# Introduction

Women's experiences of childbirth care have been reported as a global public health issue [1] as they can cause an immediate and long-term impact on the lives, well-being and health of women and their neonates. A positive birth experience can be empowering and facilitate the transition to motherhood [2], whereas a negative birth experience can cause fear of birth [3, 4] and postpartum depression [5, 6], and it can diminish maternal-neonatal bonding [7] and stunt the growth and development of a neonate [8]. Poor quality of childbirth-related care is one of the most common reasons behind negative birth experience [9]. Examples of such experiences include malpractice or negligence, lack of respect from care providers, and poor hands-on care given to women during labour and birth [10]. These are critical reasons why women avoid coming to hospitals for childbirth [11, 12]. Therefore, women's experiences need to be better understood, and gaps in respectful and supportive care need to be identified. A validated and reliable tool that can measure the women's birth experiences becomes important in such circumstances.

## Valid and reliable tools for measuring the women's birth experiences

There are existing instruments measuring women's childbirth experiences [13–18] which can guide researchers to identify and adapt an appropriate instrument for use instead of developing a new one. After reviewing the seven questionnaires measuring women's childbirth experiences, the Revised Childbirth Experience Questionnaire (CEQ2) developed by Dencker et al. (2020) was considered most suitable [18] given the widespread validation and use of the original CEQ in several low-, middle- and high-income countries [13, 19–22]. In 2020, the authors of the original CEQ updated the questionnaire by adding new questions that explored the professional support and participation-related aspects of childbirth care in addition to the original dimensions of perceptions regarding own capacity and personal safety during childbirth. This 22-item questionnaire–CEQ2 –has been validated and culturally adapted in European and Asian countries [18, 23, 24]. The CEQ2 was selected for use in India due to its broadly framed questions that allowed for easy socio-cultural contextualisation of the tool.

## Intrapartum care in India

The government of India has invested in improving maternal and child health care over the past decade, including infrastructural reforms of labour rooms [25], re-skilling of care providers in safe and sensitive care, and taking the historic decision to roll out India's first midwifery service guidelines to introduce professional midwives as an independent cadre in the Indian healthcare system. However, there is a dearth of evidence reporting women's birth experiences from India. While several qualitative studies have described Indian women's experiences of giving birth/receiving childbirth care, no standardised childbirth experience measurement questionnaire has been tested and validated in India–to the best of the authors' knowledge–to measure these experiences, nor does an Indian questionnaire exist that does so.

Testing a standardised questionnaire in India would contribute to measuring the overall birth experience of Indian women, which is an important step in identifying quality improvement areas and developing strategies for providing respectful, quality maternity care during labour and childbirth. As part of a larger project designed to implement midwife-led care in Karnataka state, India, this study aimed to test the construct validity and reliability of the *Kannada*-translated Revised Childbirth Experience Questionnaire (*Kannada*-translated CEQ2).

## Methodology

*The Revised Childbirth Experience Questionnaire (CEQ2)* developed by Dencker et al. [26], with 22 items, 19 of them structured as a 4-point Likert scale (1 = Totally disagree, 4 = Totally agree), and three items on a Visual Analogue Scale (VAS), was selected for cultural adaptation and testing in Karnataka, India. The CEQ2 has eight negatively worded items that need reverse coding before analyses. The 22 items are clustered under four domains: 1) Own capacity (8 items), 2) Perceived safety (6 items), 3) Professional support (5 items) and 4) Participation (3 items). The questionnaire has good construct validity and has been tested in European and Asian countries with moderate to significant changes in its structure. To the best of the authors' knowledge, the CEQ2 has not been translated and used in any state of India.

**Contextualising the CEQ2 for use in Karnataka, India.**   The trans-cultural questionnaire process recommended by Wild et al. [27] was followed. The authors *prepared* themselves by discussing the CEQ2 with original authors and seeking their advice. A *forward translation* of the CEQ2 was carried out from English to *Kannada* by a translation expert in the field, and *reconciliation* of all other possible translations was carried out by the Indian authors. Another independent language expert *back-translated* the questionnaire, which was reviewed by all the authors and by the author of the original CEQ2. All suggestions were incorporated. The other available translations of the CEQ2 were reviewed online to check for different translations, however no changes were made in the Kannada-translated questionnaire as our translation seemed *harmonised* with other published CEQ2 versions. *Cognitive debriefing* was carried out using a "think-aloud" technique with seven postpartum women who had recently given birth in order to test for the *Kannada*-translated CEQ2's understandability, interpretation and cultural relevance of the translation. As a result, the term "partner" was replaced with "birth companion/family member", and the term "midwife" was replaced with "care provider during labour and birth". The visual analogue scale was prepared using a pictorial representation to increase understanding of quantification among women, if required (1 dollar to 100 dollar image instead of 1–100 scale) for ensuring uniform understanding. The authors further *reviewed* the outcomes of cognitive debriefing activities to identify any further discrepancies, though none were observed. The *proof-reading* of this final prepared *Kannada*-translated CEQ2 was carried out by another translator to rule out any typological errors. It is noteworthy that all translators involved in this process were native *Kannada* speakers and were also fluent in reading and writing English.

The final version of the *Kannada*-translated CEQ2 consisted of 22 items from the original CEQ2, along with socio-demographic profiling-related questions, obstetric history-related questions, questions specific to the last birth, and questions about participants' general health (Table 1). The *Kannada*-translated CEQ2 was pilot tested with 25 women who did not participate in the actual study.

*Design, participants and data collection.* A cross-sectional survey was carried out from June 2022 to September 2022 to collect data. Postpartum women who had given birth in the six-month period before data collection (vaginally, assisted, or by Caesarean section) to at least one live newborn were invited to participate. Eight native *Kannada*-speaking research assistants, who were rigorously trained in processes of information sharing, consent taking and data collection using face-to-face interviews, collected the data. A face-to-face interview method was chosen to ensure that illiterate women–if any–had an equal opportunity to participate. To minimise response bias, all research assistants were provided with the operational definition of each key term used in 22 items of the CEQ2. They were instructed to contact the authors in case a participant's query needed information beyond the operational definitions.

**Table 1. Socio-demographic, obstetric and general health profile of the participants.**

|  | n (%) |
|---|---|
| **SOCIO-DEMOGRAPHIC PROFILE OF PARTICIPANTS** |  |
| *Age of the participants in completed years (mean, (SD))* | **25.5 (4.97)** |
| **Educational status of the participants** |  |
| *Illiterate/home schooled* | *4/2 (2.4)* |
| *Up to 10 years of formal education* | *89 (35.5)* |
| *Up to 12 years of formal education* | *75 (29.8)* |
| *Graduation or higher* | *81 (32.3)* |
| **Work status of the participants** |  |
| *Homemaker* | *174 (69.3)* |
| *Salaried employee* | *62 (24.7)* |
| *Other (contractual/project based)* | *15 (6.0)* |
| **Reported annual income of the participants' families** |  |
| *Up to 1 million Indian rupees* | *181 (72.1)* |
| *More than 1 million Indian rupees* | *70 (27.9)* |
| **Marital status of participants** |  |
| *Married* | *249 (99.2)* |
| *Widowed* | *8 (0.8)* |
| **OBSTETRIC PROFILE OF THE PARTICIPANTS** |  |
| **Participant groups based on parity** |  |
| *Primipara* | *125 (49.7)* |
| *Multipara* | *126 (50.3)* |
| **Mode of childbirth** |  |
| *Vaginal birth (with/without requiring perineal suturing)* | *162 (64.8)* |
| *Caesarean birth (planned/emergency)* | *85 (33.8)* |
| *Assisted (vacuum/forceps)* | *3 (1.2)* |
| **Self-reported duration of labour by the participants** |  |
| *Did not experience labour* | *19 (7.7)* |
| *Up to two hours of labour pain* | *39 (15.7)* |
| *>2–5 hours of labour pain* | *50 (20.2)* |
| *>5–10 hours of labour pain* | *91 (36.7)* |
| *>10 hours of labour pain* | *49 (19.7)* |
| **As reported by the participants, the birth facility belonged to** |  |
| *Private sector* | *115 (46.0)* |
| *Public sector* | *136 (54.0)* |
| **As reported by the participants, the birth facility was** |  |
| *A private specialty hospital* | *61 (24.4)* |
| *A private nursing home/family-run small hospital* | *54 (21.6)* |
| *A government medical college hospital* | *38 (14.6)* |
| *A government district hospital* | *98 (39.4)* |
| **GENERAL HEALTH PROFILE OF THE PARTICIPANTS** |  |
| **Self-perception of general health** |  |
| *Good* | *196 (78.1)* |
| *Quite Good* | *45 (17.9)* |
| *Somewhat poor* | *8 (3.2)* |
| *Poor* | *2 (0.8)* |
| *Reported being treated for anxiety in past* | *6 (2.4)* |
| *Reported being treated for depression in past* | *5 (2.0)* |

**Study site and sample size.** Women were recruited for home-based interviews from the communities around large public and private health facilities (500 or more monthly child-births) in two districts of Karnataka. A sample size of 220 participants (10 times the number of items in the CEQ2) was determined following the principles of psychometric evaluation [28], and 251 women were approached based on an expected 25% refusal to participate.

**Statistics and data analyses.** Baseline descriptive statistics (frequencies and percentage; mean and SD when required) were calculated to describe the participants who responded to the *Kannada*-translated CEQ2. Mean and SD of each item on the *Kannada*-translated CEQ2 was calculated. An exploratory Principal Component Analysis (PCA) and Confirmatory Factor Analysis (CFA) were carried out to understand the factor structure, validity, known-group validity and reliability of this questionnaire.

*Principal component analysis.* IBM SPSS 25 was used to perform PCA. A significance level of $p = 0.05$ was considered throughout the study. Kaiser-Meyer-Olkin (KMO) test of sample adequacy and Bartlett's Test of Sphericity were performed, followed by PCA using direct oblimin rotation; extracting the factors based on eigenvalues $\geq 1$, coefficients having <0.40 factor loadings were suppressed.

**Confirmatory factor analysis.** IBM SPSS AMOS 26 software was used to conduct CFA. Model fit statistics were used to assess the factor structure suggested in EFA: minimum discrepancy divided by degrees of freedom (CMIN/DF), chi squared, the Comparative Fit Index (CFI), Tucker-Lewis Index (TLI), standardised root mean square residual (SRMR), and root mean square error of approximation (RMSEA), where CMIN/DF< 5, CFI and TLI values closer to 0.95, the SRMR values closer to 0.08 and RMSEA values <0.08 demonstrate a good fit [28].

**Internal construct validity.** Item-factor, factor scale and item-scale correlations were calculated to determine the internal construct validity of the *Kannada*-translated CEQ2. Coefficients <0.2 would require deletion of that item from the scale [28].

**Internal consistency.** Cronbach's alpha coefficient (α) was calculated to determine the internal consistency. For this purpose, α would be calculated for items as clustered under all factors identified and for the total scale, where $\alpha \geq 0.70$ to $0.90$ would demonstrate good internal consistency [28].

**Known group validity.** The ability of the *Kannada*-translated CEQ2 to differentiate between different groups was carried out using the known-group validity check. Women's parity, mode of birth and the duration of labour were selected as these variables were used with original CEQ2 testing [26]. In addition, presence of a birth companion during labour was also included as a variable for known-group testing, in line with CEQ2 testing reported by Lok et al. [23]. Cohen's d effect sizes were calculated and considered trivial (<0.2), small ($\geq 0.2$–0.5), moderate ($\geq 0.5$ and $\leq 0.8$) and large (>0.8) [28].

**Ethical clearance and participants' consent.** This study was approved by the Ethical Review Board of the Foundation for Research in Health Systems (FRHS) (Referral No. 2022/MWC/01). All participating women received verbal and written information on the study and duly gave their written informed consent before responding to the CEQ2. The questionnaire was administered at a time convenient for the women and their families, minimising disruption to their daily routines. Women RAs were recruited to ensure socio-cultural acceptance of the research team by the community.

## Findings

A total of 251 women participated in the study. All invited women accepted participation. After responses were checked for completeness (none missing), all completed forms were used in the analysis. The median postnatal period in which women responded was 2.9 months (SD

**Table 2. Overview of principal component analyses for *Kannada*-translated CEQ2.**

| Factor/subscales | | Item analyses (correlations) | | | Construct Validity | | | Cronbach's alpha (α) | Range of Mean Scores (SD) of items |
|---|---|---|---|---|---|---|---|---|---|
| | | Item-total correlation range | Item-subscale correlation range | Subscale-total correlation range | eigenvalue | Variance explained (%) | Item loading range | | |
| 1 | Perceived Safety (6 items) | 0.42–0.48 | 0.82–0.94 | 0.53 | 6.4 | 29.1 | 0.75–0.91 | 0.92 | 2.10 (0.58)-2.19 (0.70) |
| 2 | Own Capacity (8 items) | 0.55–0.75 | 0.69–0.89 | 0.81 | 5.3 | 24.2 | 0.67–0.90 | 0.93 | 1.82 (0.88)-2.00 (0.98) |
| 3 | Professional Support (5 items) | 0.82–0.93 | 0.82–0.93 | 0.46 | 2.7 | 12.3 | 0.78–0.93 | 0.97 | 2.27 (0.56–2.30 (0.57) |
| 4 | Participation* (2 items) | 0.22*-0.28 | 0.91–0.93 | 0.24 | 2.2 | 10.1 | 0.90–0.93 | 0.91 | 2.57 (0.74)-2.73 (0.67) |
| **Total scale** | | **Range 0.22–0.93** | **Range 0.69–0.93** | **Range 0.24–0.81** | | **75.7%** | **-** | **0.84** | **2.14 (0.39)** |

*item 8 was deleted due to poor item-total score correlation (<0.2)

1.6). All participants were married, and a majority of the participants were homemakers (69.3%). About half of the participants were primiparous (49.7%), and 54% participants had given birth in public health facilities. About 9% of participants had experienced at least one miscarriage (ranging from 1–5 miscarriages), and 5.2% had lost an under-5-year-old child. Table 1 presents the socio-demographic, obstetric and general health profile of the participants.

**Principal component analyses.** The *Kannada*-translated CEQ2 passed the KMO test of sample adequacy (value 0.875) and was factorable as demonstrated by a significant Bartlett's test of sphericity ($\chi^2 = 4869.2$, $p = <0.001$). The PCA revealed a four-factor solution which was similar to the one proposed by Dencker et al. (2020) in their original testing [26]. However, the "Perceived safety" (items 3,16,17,18,19,22) factor explained the maximum amount of variance instead of the "Own capacity" factor (items 1,2,4,5,6,7,20,21), which was the first factor in the original CEQ2. The "Professional support" (Items 10,12,13,14,15) and "Participation" (Items 8,9,11) were the third and fourth extracted factors respectively in this study. Together, the four factors explained 75.6% of the variance. The construct validity showed that item 8 had poor correlation with the overall scale score (<0.2), and therefore the item was deleted from the scale. Table 2 presents the overview of PCA findings for the *Kannada*-translated CEQ2.

The Confirmatory Factor Analysis was carried out using the maximum likelihood modelling with the aim of justifying the compliance between the identified exploratory factors. Using the four factors with all 22 items generated a statistical model with poor fit. Therefore, the 4-factor model was re-run after removing item 8 from factor 4, "Participation", which generated a 4-factor model with an acceptable fit. Table 3 presents CFA findings from the original questionnaire, and from the 4-factor model– 22 items and 21 items respectively–as identified

**Table 3. Confirmatory factor analyses findings for two models of *Kannada*-translated CEQ2 as compared to original CEQ2 model.**

| | CMIN/DF | GFI | AGFI | Chi squared value of model fit | Degrees of freedom | P value | CFI | TLI | SRMR | RMSEA |
|---|---|---|---|---|---|---|---|---|---|---|
| *Model by the original authors* | *2.79* | *0.940* | *-* | *2348.143* | *203* | *<0.001* | *0.94* | *0.93* | *-* | *0.054* |
| *4-factor model (22 items): this study* | *3.061* | *0.825* | *0.783* | *624.375* | *204* | *<0.001* | *0.912* | *0.901* | *0.043* | *0.091* |
| *4-factor model (21 items): this study*\* | *1.334* | *0.915* | *0.891* | *240.191* | *180* | *0.002* | *0.984* | *0.986* | *0.042* | *0.037* |

*retained model with good fit

**Table 4. Differences in subscale scores and overall score of *Kannada*-translated CEQ2 by different groups.**

| | n | Perceived safety (mean, SD) | Own capacity (mean, SD) | Professional support (mean, SD) | Participation (mean, SD) | Mean Kannada-translated CEQ2 score (mean, SD) |
|---|---|---|---|---|---|---|
| *Primipara* | 125 | 2.10 (0.60) | 1.71 (0.72) | 2.23 (0.51) | 2.68 (0.64) | 2.04 (0.35) |
| *Multipara* | 126 | 2.23 (0.51) | 2.10 (0.79) | 2.34 (0.51) | 2.62 (0.59) | 2.24 (0.39) |
| ***Unadjusted p value*** | | **0.03** | **<0.001** | 0.07 | 0.65 | **<0.001** |
| *Gave birth vaginally (spontaneous/ assisted)* | 166 | 2.21 (0.53) | 1.97 (0.78) | 2.27 (0.47) | 2.64 (0.57) | 2.17 (0.38) |
| *Gave birth through Caesarean Section (planned/emergency)* | 85 | 2.08 (0.61) | 1.77 (0.75) | 2.32 (0.56) | 2.68 (0.71) | 2.07 (0.39) |
| ***Unadjusted p value*** | | 0.31 | **0.04** | 0.69 | 0.39 | 0.07 |
| *Total duration of labour < 12 hours* | 200 | 2.17 (0.52) | 1.92 (0.77) | 2.32 (0.53) | 2.66 (0.62) | 2.16 (0.39) |
| *Total duration of labour ≥ 12 hours* | 51 | 2.16 (0.52) | 1.86 (0.83) | 2.17 (0.38) | 2.60 (0.64) | 2.08 (0.38) |
| ***Unadjusted p value*** | | 0.59 | 0.46 | 0.07 | 0.64 | 0.23 |
| *Birth companion for as long she wanted* | 46 | 2.03 (0.59) | 2.18 (0.81) | 2.22 (0.43) | 2.65 (0.60) | 2.19 (0.36) |
| *Birth companion not allowed* | 205 | 2.19 (0.56) | 1.84 (0.75) | 2.31 (0.53) | 2.65 (0.63) | 2.13 (0.39) |
| ***Unadjusted p value*** | | 0.102 | **0.004** | 0.46 | 0.30 | 0.16 |

in this study, whereas Table 4 presents the differences in subscale scores and overall average score on 21-items *Kannada*-CEQ2 by the participants divided based on their socio-demographic and work-related characteristics.

## Discussion

This study aimed to culturally adapt and evaluate the psychometric properties of the *Kannada*-translated CEQ2; to the best of our knowledge, this is the first study to do so. Based on exploratory and confirmatory factor analyses findings in this study, the 21-item *Kannada*-translated CEQ2 was found to be a reliable and valid questionnaire for use within *Kannada*-speaking populations. The *Kannada*-translated questionnaire had good Cronbach alpha scores for all subscales and overall scales (≥ 0.84).

The *Kannada*-translated CEQ2 had the same factor structure as reported for the original CEQ2 [26]. This was in line with the findings from other studies [18, 24] but contrasted with the study done in Hong Kong [23], where nine items had to be removed from the original 22 items. This may be because the respondents from our study were literate, professional women (although many were currently not working and identified as homemakers) who could relate to all items of the CEQ2, as evidenced by the face validation exercise. It is noteworthy that the fourth factor in the *Kannada*-translated CEQ2, "Participation", had lower subscale—total scale correlations during construct validity testing when compared to the other three factors, which could be a reflection on how Indian women prioritise their own involvement in birthing care planning and implementation as compared to "giving up control". Some qualitative studies from India have suggested that the women voluntarily gave up control because they trusted the expertise of the care providers in keeping both the neonate and mother safe [29–31]. However, there is a need to further explore this aspect through other studies, especially because the Government of India has committed to promoting women-centred, respectful maternity care in the country, and promoting the women's participation is critical in realising this vision [32].

Results from this study showed that, for the most part, there were no differences in women's scores across known groups (primi/multipara; vaginal/CS births; presence/absence of birth companions; and less than/more than 12 hours of labour pains) except for the differences

having logical explanations. For example, multiparous women were more likely to have positive experiences of "perceived safety" (*p* value 0.03) and "Own capacity" (*p* value <0.001) compared to primiparous women, interpreted in light of the fact that their previous birth experiences had readied them for what to expect. This was in line with other studies using the CEQ or CEQ2 [21, 23, 26]. Women who gave birth vaginally scored higher on "Own capacity" (*p* value 0.04) compared to women who had birthed through Caesarean Section; this was in line with other studies validating the CEQ or CEQ2 [21, 23, 26]. Women who had a birth companion with them for as long as they wished scored higher on "Own capacity" (*p* value 0.004). It could be that the birth companion's emotional support boosted the women's confidence during birth, a phenomenon reported in some recent studies from India [33, 34].

Our study–despite being at a community level and towards the end of the acute phase of COVID-19 –had a 100% response rate. This could be because postnatal women in India may not have the opportunity to talk openly about their birth experiences, due to the cultural beliefs that disclosing details of birth and babies brings evil eyes [35], which leaves a critical unmet need, known to impact women's transition into motherhood as well as their mental health [36–38].

Being a short questionnaire using self-assessment, this questionnaire is an easy, time- and cost-saving tool to administer for measuring the quality of childbirth-related care from the women's perspective. Additionally, this tool provides a unique opportunity to measure women's childbirth experiences that are known to be a sensitive indicator of quality of childbirth care and can serve as a valid instrument to record women's birth experiences and comparisons across midwife-led care units and care received in traditional labour rooms.

## Strengths and limitations

To the best of the authors' knowledge, this study is the first of its kind validating the CEQ2 for use in India among *Kannada*-speaking women. The study adopted a rigorous methodology to improve data quality. Data were collected with an adequate number of participants by experienced researchers who were further trained for implementing this study. The data quality was closely monitored by the researchers. Research teams carried printouts of the definitions of key terms used in each item of the CEQ2 and used those to answer any queries during data collection to mitigate bias during data collection. However, participants' responses remain subjective and may have intrinsic response bias, which is an inherent constraint to all surveys seeking human response. The CEQ2 has been validated for *Kannada*-speaking women and should be re-validated before use in other regions of India. While the CEQ2 is a multidimensional tool validated to capture the multifaceted nature of birth experiences, there may be some aspects of birth experience that could be beyond quantification and may not have been captured.

## Conclusion

The *Kannada*-translated CEQ2 validated through this study has good internal consistency and construct validity and can be used for ongoing measurement of *Kannada-speaking* women's birth experiences and for providing the evidence to further improve quality of childbirth care in India.

## Acknowledgments

We are thankful to the women and their families who consented to participate in this study.

## Author Contributions

**Conceptualization:** Paridhi Jha, Bharati Sharma, Kerstin Erlandsson, Malin Bogren.

**Data curation:** Paridhi Jha, Bharati Sharma.

**Formal analysis:** Paridhi Jha, Vikas Kumar Jha, Bharati Sharma, Ajeya Jha.

**Funding acquisition:** Malin Bogren.

**Methodology:** Paridhi Jha, Vikas Kumar Jha, Bharati Sharma, Ajeya Jha, Kerstin Erlandsson, Malin Bogren.

**Project administration:** Malin Bogren.

**Validation:** Paridhi Jha, Bharati Sharma, Ajeya Jha, Kerstin Erlandsson, Malin Bogren.

**Visualization:** Bharati Sharma, Kerstin Erlandsson, Malin Bogren.

**Writing – original draft:** Paridhi Jha.

**Writing – review & editing:** Vikas Kumar Jha, Bharati Sharma, Ajeya Jha, Kerstin Erlandsson, Malin Bogren.

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
